# Investigating Gender-based violence against internally displaced women in Debre Berhan, Central Ethiopia: A mixed-methods study using the socio-ecological framework

Kindie Mitiku[1]*, Sisay Shiwasinad[2], Solomon Shiferaw[3]

1 School of Public Health, College of Health Sciences, Debre Tabor University, 2 Department of Pediatric and Child Health Nursing, College of Health Sciences, Debre Berhan University, 3 School of Public Health, College of Health Sciences, Addis Ababa University

* mitikukindie@gmail.com

## Abstract

### Background

Gender-based violence (GBV) is a major health problem affecting displaced populations disproportionately. However, limited research existed on the prevalence, barriers, and facilitators for survivors seeking care.

### Objective

This study aims to estimate the prevalence of GBV and investigate the barriers and facilitators influencing survivors' access to care.

### Methods

A mixed-methods cross-sectional study was conducted in 2024 involving 1,863 women. Women were recruited through random sampling. The qualitative component included five NGO workers and eleven GBV survivors, who were selected purposively. Quantitative data were collected using the Assessment Screen to Identify Survivors Toolkit. The qualitative data were analysed thematically with Atlas Ti 8, guided by the socio-ecological framework.

### Results

Nearly one-third (31%) of women experienced GBV, with 25.2% of them facing it in the past year. The most common types of violence were threats of violence (32.1%), physical violence (25.8%), forced marriage (19.1%), and sexual violence (10.0%). Nearly 80% of GBV incidents took place in IDP camps, mainly perpetrated by intimate partners and family members. Barriers to seeking GBV services at the individual level included self-isolation, reluctance to disclose survivor status, and lack

**Data availability statement:** All relevant data are within the manuscript and its Supporting information files in the form of SPSS data set.

**Funding:** Authors who received each award: KMK Grant numbers awarded the author:09/2024 The full name of each funder's. ST.PAUL INSTITUTE FOR REPRODUCTIVE HEALTH AND RIGHTS URL of each funder website:https://www.spirhr.org/ Funders role: The funder played a role in the study design, data collection, and analysis but did not provide funding for manuscript preparation and publication

**Competing interests:** The authors have declared that no competing interests exist.

of awareness. Community-level restrictions comprised social stigma, gossip, and inadequate social support, while institutional challenges involved budget constraints and a lack of confidentiality. Structural barriers included camp overcrowding, insecurity, and mistrust in the justice system. Self-efficacy acted as an individual-level enabler for survivors to seek care. Enablers at the institutional level included support from NGOs, access to secure housing, and availability of a one-stop centre. Access to community-based GBV workers was viewed as a crucial community-level facilitator for survivors seeking care.

## Conclusions

GBV is widespread among internally displaced women, particularly in camps. Despite the presence of some facilitators, GBV survivors encounter numerous barriers at all levels of the socio-ecological framework. Overcoming these barriers requires comprehensive and coordinated efforts. Key strategies include increasing awareness of the available GBV services, reducing community stigma, building supportive networks, safeguarding survivors' privacy, decreasing overcrowding in camps, enhancing security measures, and rebuilding trust in justice systems.

---

## Background

Gender-based violence (GBV) is a severe human rights violation and a life-threatening crisis that affects people from all regions [1]. The United Nations High Commissioner for Refugees (UNHCR) defines GBV as: "a comprehensive term for any detrimental act inflicted against an individual's consent, predicated on gender disparities between males and females. It encompasses behaviours that cause bodily, sexual, or psychological harm or suffering, threats of such actions, coercion, and other forms of liberty deprivation. These behaviours can take place in either public or private situations"[2].

In the context of displacement, GBV includes any physical, psychological, sexual or economic act that harms women's safety and dignity due to displacement. In displacement situations, GBV can take many forms, such as sexual abuse and exploitation in exchange for basic survival needs, early or forced marriage, denial of access to essential services, violence by intimate partners, state and non-state actors, and workers from humanitarian organisations [2,3].

According to the World Health Organisation (WHO), approximately one-third of women worldwide have experienced physical or sexual assault in their lifetime, indicating that GBV is a global public health concern [1]. Despite being a global public health concern, GBV disproportionately impacts displaced populations. Displaced populations, such as refugees and internally displaced persons(IDPs), are more vulnerable to the various kinds of GBV than non-displaced populations due to a lack of social networks [4,5], institutional breakdown [6], and loss of access to vital services

[7]. Even though GBV is a global issue, the unique experiences of refugees and IDPs remain barely investigated, particularly in regions with large-scale displacement circumstances [8,9].

IDPs are "persons or groups of persons who have been forced or obliged to flee or to leave their homes or places of habitual residence, in particular as a result of or to avoid the effects of armed conflict, situations of generalized violence, violations of human rights or natural or human-made disasters, and who have not crossed an internationally recognized state border" [10].

Ethiopia has experienced a significant increase in IDPs, with over 3.3 million reported in 2024, primarily due to armed conflicts and inter-ethnic clashes (68.7%), drought (16.5%), and other factors [11]. Debre Berhan, which is located in the central part of Ethiopia, has evolved as a popular place for IDPs. Nearly 37,000 IDPs in the North Shewa zone, the majority of them over 20,000 in Debre Berhan City, were reported as of 2022 [12].

Debre Berhan has been an important place for IDPs for several reasons. It is close to Addis Ababa, Ethiopia's capital. Because of its central location, the city serves as an ideal transit and settlement destination for displaced persons. Furthermore, Debere Berhan had remained reasonably safe and stable until the beginning of the conflicts in the Amhara region [13], making it an attractive destination for displaced persons.

However, the country's broader political, social, and economic difficulties, combined with the camp conditions, may make IDPs vulnerable to GBV. The deep-rooted patriarchal norms and gender inequity, and lack of financial autonomy make them more prone to GBV [14–16]. The ongoing unrest in the country's Amhara area may render them even more exposed to the violence [5,17].

The situation of displaced individuals in Debre Berhan appears to be much different from those in other IDP camps throughout Ethiopia. Settlements in Debre Berhan are primarily peri-urban and informal, operated and maintained by the city with little assistance from humanitarian agencies. People in these IDP camps either live in shared homes or tiny tents [18]. The lack of adequate food and other basics makes things difficult [19]. Furthermore, the already overburdened healthcare system and safety concerns [20] may raise the risk of GBV for these vulnerable people.

The available evidence in Africa shows that GBV is a significant issue, with a meta-analysis revealing that 48.2% of refugees and IDPs experienced it, while IDPs had a prevalence of 36.24% [5]. Studies in East Africa have also found that GBV is a severe public health concern among IDPs, with a 37.9% prevalence in northwest Ethiopia [4], 18.5% in Somalia [21] and 51.7% in Northern Uganda [22].

Displaced women face numerous barriers to accessing GBV and sexual reproductive health (SRH) services, including fear of stigma, humiliation, isolation, and blame [21,23–25]. They lack understanding of available services [24–26] and their rights [21,25], which deters them from reporting abuse. Community-level challenges include social stigma [23,25], normalisation of violence [24,25], and preference for community mediators [24]. Institutional barriers include a lack of anonymity [23,25] and financial problems [26]. Structural-level constraints like insecurity [23] and a lack of trust in the justice system [21,25] also impede survivors' access to care and support.

Despite the aforementioned barriers to receiving GBV services, there are also various facilitators for survivors to get assistance and care. These include self-perceived high severity of the violence [23]; availability of awareness creation activities [26,27]; and community health volunteers who provide health education for refugees to increase awareness [23,24].

Understanding the contextual barriers and facilitators that influence GBV survivors' access to care is crucial for establishing a more effective GBV response in displacement contexts. The few previous studies in Ethiopia have focused on assessing the prevalence of GBV in war-affected areas [28,29]. For instance, Dellie et al reported GBV prevalence of 39% in the conflict-affected areas of Northeast Amhara [28]. Similarly, Asefa EY et al reported GBV prevalence of 58% in the post-war districts of North Shewa zone [29]. Some studies have also focused on the overall SRH of refugees and IDPs [30,31]. Only one study looked at the prevalence of GBV among IDPs in Northern Ethiopia. This study reported GBV prevalence of 37.9% [4]. However, the studies among IDPs in Northern Ethiopia [4] and war-affected areas [28,29] did

not employ an analytical framework that captured the multi-level influences on GBV service access. To the researchers' knowledge, no study has used the socio-ecological framework to determine the facilitators and barriers to treatment-seeking behaviour of GBV survivors among IDPs in Ethiopia, including Debre Berhan. Furthermore, given the presence of vulnerabilities to GBV among IDPs in Debre Berhan IDP camps, the prevalence of GBV and the type of perpetrators remain unknown.

Addressing the burden of GBV in Ethiopia, particularly in the refugee settings, needs a comprehensive strategy targeting institutional barriers and the socio-political factors affecting service delivery [32]. Furthermore, the complex nature of GBV requires multi-level solutions that are culturally sensitive and responsive to the specific problems encountered by displaced populations [24,31]. This, in turn, indicates a need to study GBV in broader contexts using a socio-ecological framework.

This study aims to overcome the aforementioned research gaps by determining the prevalence of GBV and identifying the barriers and enablers of access to GBV care among displaced women in Debre Berhan, Ethiopia. This study, guided by the socio-ecological framework, explored the individual, communal, institutional, and structural factors influencing GBV survivors' access to care. Thus, the study's results will provide significant information to guide policy and practice, enabling more targeted actions to reduce GBV and strengthen care and support services for displaced women in Ethiopia and in other countries with similar settings.

## Methods

### Study area and period

The study was conducted at three internally displaced camps in Debre Berhan, central Ethiopia. The camps are named China, Woynishet, and Bakelo, with an estimated total population of 23,093. Nearly half (50.2%) of the displaced population are female, and almost half reside in the China camp, the largest of the three. Women aged 15–49 comprise approximately one-quarter of the total refugee population across the camps. The study was conducted between January 30 and February 29, 2024.

### Study design

A parallel mixed-methods cross-sectional study was conducted using both qualitative and quantitative components. The two approaches addressed different objectives. The quantitative component estimated GBV prevalence. On the other hand, the qualitative component explored barriers and facilitators for accessing care and support among survivors of GBV. Though the two approaches had distinct goals, we gathered data simultaneously and combined the findings in the discussion to present a holistic perspective of GBV among IDP settlements.

### Study participants

The study population for the quantitative component included women of reproductive age (15–49 years). For the qualitative component, participants consisted of survivors of physical and/or sexual violence, as well as key informants working on GBV, such as mental health and psychosocial support (MPSS) workers, social workers, and case managers.

### Sample size and sampling procedure

The study included 1,863 women of reproductive age from three camps, chosen by systematic sampling. An initial census was conducted to determine eligible women and create a sampling frame. We completed the whole enumeration with the assistance of camp coordinators and community focal personnel. Many families in each camp shared large tents and established separate demarcated zones with their supplies, such as blankets and plastic sheets. For census reasons, we considered each demarcated area to be a household. All reproductive-age women (15–49 years old) who lived in the camps for at least six months were identified and included in the sample frame.

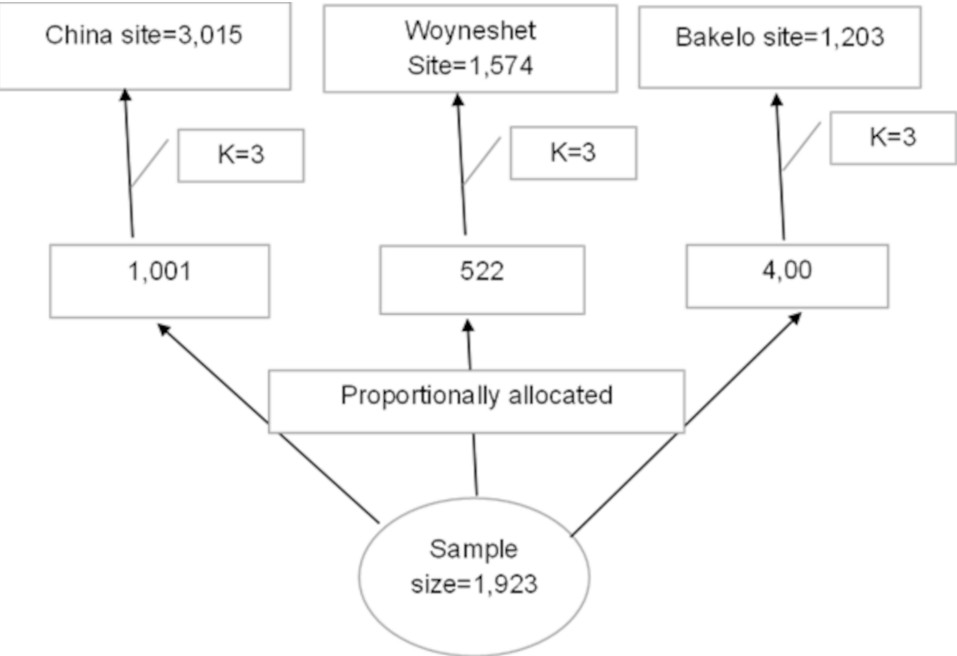

**Fig 1. Sampling procedure for the study gender-based violence among internally displaced women in Debre Berhan, Central Ethiopia,2024.**

The sample size was proportionally assigned to each camp. The sampling fraction was calculated by dividing the number of eligible women in each camp by the allocated sample size (Fig 1). For the qualitative study, the sample size was determined based on information saturation, and 16 participants (11 GBV survivors and 5 key informants) were included. The study participants were chosen purposefully for the qualitative component.

### Eligibility criteria

The study excluded women of reproductive age who were seriously ill or mentally incapable of responding. Survivors of physical and/or sexual violence were included, representing diverse age groups, educational backgrounds, and experiences with service use. Key informants working on case management, counselling, and social support were included to understand barriers and facilitators affecting survivors' access to GBV services.

### Data collection tools and procedures

For the quantitative component, a structured questionnaire was developed based on a review of the literature on GBV in humanitarian settings [4,5,25,27,33]. The questionnaire was then pretested on 5% of the sampled population. Those who participated in the pretest were excluded from the actual data collection. Following the pretest, minor changes were made to the questionnaire. The questionnaire included respondents' socio-demographic information, types of GBV, when they occurred, and the perpetrators of the violence (S1 File). The quantitative data collection takes around 30–40 minutes.

For the qualitative component, semi-structured interview guides were developed after reviewing qualitative studies conducted in humanitarian settings [23,24]. The guiding questions for GBV survivors (S2 File) and key informants (S3 File) are provided as supplementary files.

GBV survivors were identified and recruited through two mechanisms. First, we collaborated with Marie Stops International and the Ethiopian Orthodox Church Development and Inter-Church Aid Commission to recruit

survivors of sexual violence who were enrolled for care and support. Those organisations helped the data collectors recruit sexual violence survivors who were already seeking therapy and aid from them. Each data collector contacted one GBV worker from each organisation and informed them about the eligibility criteria for the study. Then, GBV workers from each organisation recommended those sexual violence survivors who expressed an interest in participating.

Second, during the quantitative data collection, survivors of physical and/or sexual violence were asked whether they would want to be contacted for an optional follow-up interview. Those who agreed were assigned to a second team of qualitative data collectors, with no overlap between the teams to maintain privacy. Survivors were included in the qualitative research if they were 15 or older, had directly experienced GBV, and gave informed consent.

In-depth interviews with GBV survivors were conducted in women's and girls' safe spaces, with the option for participants to choose a private location if they felt uncomfortable. Interviews with key informants, including case managers, counsellors, and social support workers, were held in rooms selected by the participants to avoid interruptions from clients or customers. All interviewees were audio recorded. The interviews were conducted in Amharic and lasted up to one hour.

### Research team composition and relationship with participants

The research team included seven data collectors, two supervisors, and three principal investigators. Five of the data collectors were responsible for gathering quantitative data. The remaining two collected qualitative data. One supervisor oversaw the qualitative data collection, while the other supervised the quantitative component.

All quantitative data collectors were women. The qualitative data collectors were both men and women. The female qualitative data collector interviewed female participants, while the male collected data from key informants. The qualitative data collectors had a minimum of a Bachelor of Science degree and previous experience in collecting qualitative data. The quantitative data collectors had at least a diploma level of education.

The principal investigators were male university instructors with health science qualifications. The data collectors, supervisors, and investigators had no prior contact with the research participants.

### Measurement of GBV

The study used the ASIST-GBV questionnaire to determine the prevalence of GBV. The questionnaire is a rapid screening tool and has been validated in a humanitarian setting. It has been validated in refugee communities in Ethiopia and Colombia [33]. It has also been used in humanitarian settings in Northern Ethiopia [4].

The ASIST-GBV questionnaire has good psychometric properties and reliability, as reported from Ethiopian and Colombian research findings. The tool's internal consistency was sufficient in both nations, with Cronbach's alpha of 0.72 in Colombia and 0.77 in Ethiopia [33].

The ASIST-GBV includes seven questions that assess the many aspects of GBV, such as threats of violence, physical violence, forced sex, forced pregnancy, forced marriage, sexual exploitation, and forced abortion. We analysed GBV using the six items, as one of the seven items (forced abortion) was omitted in the Ethiopian ASIST-GBV. Table 1 depicts the ASIST-GBV components examined in our investigation. Each of the six items is coded as "no" (0) or "yes" (1). Lifetime GBV was defined as one or more "yes" responses to any of the six ASIST-GBV questions at any point in the respondent's lifetime, and recent GBV as one or more "yes" responses in the 12 months before the interview.

### Quantitative data analysis

Quantitative data were collected using an Android application (Kobo Toolbox) and exported to SPSS version 22 for analysis. Descriptive statistics, including frequencies, percentages, and medians, were employed to summarise the socio-demographic data and the prevalence of GBV. The findings were displayed in tabular format.

**Table 1. ASIST-GBV screening questions.**

| No. | Items | Response | Did this happen in the past year? For 'Yes' responses only |
|---|---|---|---|
| 1 | Have you ever been threatened with physical or sexual violence by someone in your home or outside of your home? | 1.Yes 2.No | 1.Yes 2.No |
| 2 | Have you ever been hit, punched, kicked, slapped, choked, hurt with a weapon, or otherwise physically hurt by someone in your home or outside of your house? | 1.Yes 2.No | 1.Yes 2.No |
| 3 | Have you ever been forced to have sex against your will? | 1.Yes 2.No | 1.Yes 2.No |
| 4. | Have you ever been forced to have sex to be able to eat, have shelter, or have sex for essential services (such as protection or school) because you or someone in your family would be in physical danger if you refused? | 1.Yes 2.No | 1.Yes 2.No |
| 5 | Have you ever been physically forced or made to feel that you had to become pregnant against your will? | 1.Yes 2.No | 1.Yes 2.No |
| 6 | Have you ever been coerced or forced into marriage? | 1.Yes 2.No | 1.Yes 2.No |

## Qualitative data analysis

Thematic analysis was carried out utilising both deductive and inductive methods. The socioecological framework guided the deductive analysis. The framework comprises four levels: individual, communal, organisational, and structural. These four levels affect GBV survivors' access to care. Individual-level influences include personal traits such as beliefs, knowledge, prior experiences, and coping mechanisms. Norms, attitudes, and social support are communal-level factors influencing GBV survivors' access to care. Institutional components include organisational support, referral networks, and healthcare quality. Legal safeguards, security concerns, and the general camp conditions are structural-level factors influencing GBV survivors' access to care. Inductive coding was also employed to discover new codes and sub-themes.

The lead investigators, KM and SS, transcribed and translated the Amharic transcripts from Amharic to English with the support of language experts. They then imported the translated texts into ATLAS. ti 8 for further analysis. A hybrid technique for code-book construction was used, with KM and SS building a code-book based on the socio-ecological framework.

Open coding was also applied to investigate 20% of the transcripts, uncovering emergent codes and sub-themes. The revised code-book (S4 File) was applied to the complete dataset, with frequent discussions to ensure consistency. The core analysis was deductive, but the inductive coding process found new sub-themes that had not been part of the basic socio-ecological framework. For example, at the community level, sub-themes such as "community GBV workers" and "community support mechanisms" were developed. At the structural level, sub-themes like "security concerns" and "camp conditions" emerged. The data were grouped into 11 sub-themes, classified under the four core themes generated from the socio-ecological framework. Narrative descriptions and direct quotes from participants were utilised to highlight the sub-themes. The research follows the Consolidated Criteria for Reporting Qualitative Study (COREQ) guidelines for rigour in reporting, with a comprehensive checklist of COREQ items and page references presented as a supplementary file (S5 File).

## Ethical approval

Ethical approval letter for the study was obtained from Debre Berhan University (Ref. No. 01/02/2016 and Protocol No. 196) (S6 File). For study participants under the age of 18, parental or guardian consent was waived by the research ethics committee. This is because GBV is a sensitive subject, and involving parents or guardians may inadvertently cause harm or risks to GBV survivors. Study participants were given thorough information about the study's potential risks and

benefits. To maintain the privacy of participants, verbal consent was obtained rather than written authorisation. We chose this technique specifically to ensure participant safety. In the presence of an impartial witness who verified the participant's consent, the research team recorded the participant's verbal permission. The witness's confirmation and the participant's agreement were recorded in the research logs. Data collectors strictly followed the trauma-informed approach throughout the data collection process to ensure that no harm was done to GBV survivors. To preserve participant privacy, all data was rigorously anonymised, with personal identifiers erased throughout data collection, processing, and reporting. A cooperation letter was obtained from the Zonal IDP Coordination Office.

## Results

The study included 1,863 women of reproductive age, with a response rate of 96.9%. Most of the women were between 20 and 29 years old, and over half could not read or write. Nearly half of the respondents were current contraception users, whereas one in ten used long-acting reversible contraceptives (LARCs) (Table 2).

**Table 2. Background characteristics of respondents aged 15–49 years old among internally displaced people in the Debre Berhan IDP sites,2024.**

| Variables | % | N |
|---|---|---|
| Age in years (median age) | 30 | |
| 15-19 | 13.8 | 257 |
| 20-29 | 30.4 | 566 |
| 30-39 | 32.7 | 610 |
| 40-49 | 23.1 | 430 |
| Education | | |
| Unable to read and write | 65.6 | 1222 |
| Able to read and write | 5.7 | 107 |
| Primary (from grade 1–8) | 25.0 | 465 |
| Secondary and above (≥ grade 9) | 3.7 | 69 |
| Marital status | | |
| Married | 65.1 | 1212 |
| Single | 14.4 | 269 |
| Separated | 9.9 | 184 |
| Divorced | 10.6 | 198 |
| No of children | | |
| 0 | 19.8 | 368 |
| 1-2 | 30.1 | 560 |
| 3-5 | 38.2 | 711 |
| 6+ | 12.0 | 224 |
| Currently using contraception | | |
| Yes | 44.3 | 826 |
| No | 55.7 | 1037 |
| Currently using LARCs | | |
| Yes | 12.5 | 1631 |
| No | 87.5 | 232 |
| Total | | 1863 |

**Table 3. Types of GBVs reported in the year preceding the interview among internally displaced reproductive age women in the Debre Berhan IDP sites, 2024.**

| Type of Violence | Frequency(N) | Percentage (%) | 95% CI (%) |
|---|---|---|---|
| Threatened | 151 | 32.1 | 27.8–36.4 |
| Physically hurt | 121 | 25.8 | 21.8–29.8 |
| Forced marriage | 90 | 19.1 | 15.6–22.6 |
| Forced sex | 47 | 10.0 | 7.3–12.7 |
| Forced pregnancy | 40 | 8.5 | 6.0–11.1 |
| Sexual exploitation | 23 | 4.8 | 2.9–6.7 |
| Total | 470 | 100.0 | |

## Prevalence of gender-based violence

The study found that 31% of the surveyed women (95% CI: 28.9%–33.1%) had experienced at least one form of GBV in their lifetime. In the year before the interview, 470 (25.2%) of women (95% (CI: 23.3%–27.3%) reported experiencing GBV. In the year preceding the interview, over one-third reported threats of violence and nearly one-quarter reported physical harm. Additionally, one in ten women reported forced sex during the same period (Table 3).

## GBV during the respondents' migration journeys

Table 4 shows the GBV cases reported by internally displaced women at different migration phases. The bulk of GBV cases (73%; 95% CI: 69.4%– 76.6%) occurred in the camps. Of the total GBV cases, 3.8% (95% CI: 2.3% – 5.3% were experienced while travelling. Women faced GBV on several occasions throughout their journey. For example, 1.9% of women (95% CI: 0.8% – 3.0%) reported experiencing GBV in their homes and during travel. Although not statistically significant, 0.3% of women (95% CI: 0% – 0.7%) reported GBV while in transit and the camp.

## Perpetrators of GBV in the last 12 months

Table 5 depicts the offenders of GBV encountered by women in the 12 months preceding the interview. Intimate partners and/or family members were the top culprits. Intimate partners and/or family members were responsible for almost 80% of the instances of GBV reported in the past 12 months preceding the interview. One in ten women (95% CI: 7.4–13.0) also indicated that they had encountered GBV from security or military personnel.

## Background characteristics of qualitative study participants

The qualitative research recruited 16 individuals, including 5 key informants, 5 sexual violence survivors, and 6 physical violence survivors. The sexual violence survivors were aged between 15 and 23 years, whereas the physical violence survivors varied from 17 to 39 years.

**Table 4. GBV among internally displaced reproductive-aged women across the span of their migration experience in Debre Berhan, Ethiopia, 2024.**

| The place where the violence occurred | Frequency(N) | Percentage (%) | 95% CI (%) |
|---|---|---|---|
| Camp | 421 | 73.0 | 69.4–76.6 |
| Home | 121 | 21.0 | 17.7–24.3 |
| Transit/Travel | 22 | 3.8 | 2.3–5.3 |
| Home and transit | 11 | 1.9 | 0.8–3.0 |
| Camp and transit | 2 | 0.3 | 0.0–0.7 |
| Total | 577 | 100.0 | |

**Table 5. Perpetrators of GBV (experienced in the last 12 months preceding the interview) among internally displaced reproductive-aged women in Debre Berhan IDP sites,2024.**

| Perpetrators | Frequency(N) | Percentage (%) | 95% CI (%) |
|---|---|---|---|
| Only intimate partners | 212 | 45.1 | 40.6-49.6 |
| Only family | 104 | 21.9 | 18.2-25.6 |
| Family and intimate partner | 56 | 11.9 | 9.0-14.8 |
| Security or military personnel* | 48 | 10.2 | 7.4-13.0 |
| Neighbors | 48 | 10.2 | 7.4-13.0 |
| Others** | 3 | 0.7 | 0.0-1.4 |
| Total | 470 | 100 | |

*Policies, soldiers, and guards.

**NGO workers and unidentified individuals.

In terms of education, none of the survivors of sexual or physical violence had attended school beyond the elementary level. The key informants were aged 25–52 years and had a minimum of two years' work experience (Table 6). We established eleven sub-themes, categorised under the overarching themes of the four levels of the socio-ecological framework. Fig 2 depicts the categorisation of each sub-theme according to the four levels of the socio-ecological framework. We outline the barriers and facilitators for each level and sub-theme, highlighting real participant quotes to highlight these ideas.

### Individual-level barriers and enablers to accessing GBV Services

**Emotional and psychological responses.** Survivors typically feel self-blame, believing they are responsible for the violence. One victim commented, *"She may not go to the court if it is her fault" (Physical violence survivor 6).* Fear of stigma and gossip further deters survivors' help-seeking behaviour. A service provider observed, *"Even if we referred her to the hospital or legal services, she might refuse because she believes others are pointing fingers at her" (Key informant 1).* Survivors also fear reprisal, with one stating, *"She fears the perpetrator may cause another serious injury if she goes to the court" (Physical violence survivor 6).* Additionally, anxieties about family dissolution hinder survivors from receiving therapy. As one victim noted, *"If she has a concern about her marriage, she fears and doesn't expose herself" (Physical violence survivor 2).*

**Cognitive and awareness challenges.** Self-efficacy, acceptance of violence, and a lack of knowledge have a substantial influence on survivors' involvement in the available programs. Confidence promotes access, as one victim stated, *"With confidence in myself, I decided to go to the health facility" (Sexual violence survivor 1).* In contrast, tolerating violence becomes a barrier, with some perceiving GBV as normal. One participant noted, *"Women themselves say, why they [police/prosecutors] asked him [the perpetrator], as far as we love him" (Physical violence survivor 4).* Many survivors remain uninformed of available services, as demonstrated by one's statement: *"I haven't even seen or heard that healthcare is provided to GBV survivors" (Physical violence survivor 7).*

**Table 6. Background characteristics of respondents participating in the qualitative study in Debre Berhan, Ethiopia, 2024.**

| Types of participants | Numbers | Age ranges(years) | Education status | Sex | Years of professional experience for key informants |
|---|---|---|---|---|---|
| Sexual violence survivors | 5 | 15-23 | Not more than a primary level of education | All female | NA |
| Physical violence survivors | 6 | 17-39 | Not more than a primary level of education | All female | NA |
| Key informants | 5 | 25-52 | 4 of them BSc. and one MSc. | 1 male, 4 females | 2 −28 years |

NA = Not Applicable.

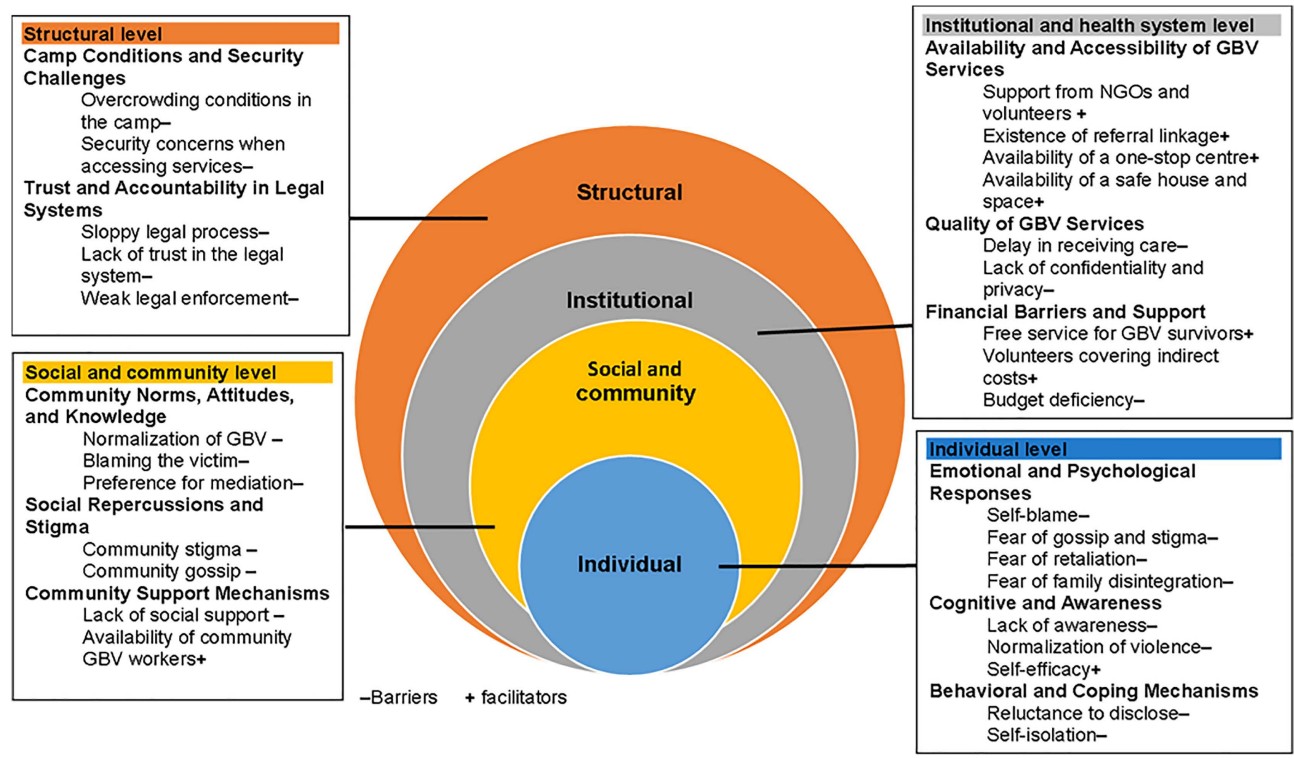

**Fig 2. Socio-ecological framework showing the barriers and facilitators for accessing GBV services among GBV survivors in Deber Berhan, central Ethiopia,2024.**

**Behavioural and coping mechanisms.** Reluctance to disclose and self-isolation impede survivors' access to help. GBV survivors opt to stay silent due to fear or stigma. One GBV survivor mentioned, *"Most of the time, we, the survivors, don't make the condition clear to the concerned bodies" (Sexual violence survivor 1).* Self-isolation is another usual coping mechanism. As one GBV survivor described, *"Because I don't want to integrate with other people, I didn't tell them about my condition" (Physical violence survivor 4).* These attitudes, caused by fear and mistrust, discourage GBV survivors from getting the appropriate medical care.

## Social and community level barriers and enablers to accessing GBV Services

**Community norms, attitudes, and knowledge.** Community norms greatly contribute to the acceptance of GBV as normal. Harmful practices like female genital mutilation (FGM) are seen as accepted norms. One key informant stated, "*Most of the community believes that female genital mutilation is normal. It is normal in their homeland" (Key informant 3).* Similarly, abuse in marriages is frequently misconstrued as devotion: *"If her husband is always beating her, she may consider he is expressing his love to her" (Key informant 1).* Victim blaming also continues, with survivors generally deemed accountable for the abuse: *"They say it is your obligation...even if I was the victim, they blamed me" (Sexual violence survivor 4).* Many presume survivors instigate violence: *"Most of the time, people assume that it is she [GBV survivor] who ignited the violence" (Key informant 3).*

Survivors generally choose informal mediation over legal action, particularly when the abuse appears "minor." A survivor added, *"If both parties agree, mediators can help resolve the situation. Otherwise, she may seek legal help" (Physical violence survivor 2).* However, inadequate knowledge of GBV perpetuates cycles of GBV and discourages

survivors from seeking help. A key informant noted, *"Women may conceal their grief...due to low awareness of GBV"* *(Key informant 2).*

**Social repercussions and stigma.** Community stigma isolates GBV survivors, who typically fear being rejected: *"They fear isolation...they consider the survivor as garbage" (Sexual violence survivor 1).* Fear of condemnation hinders survivors from getting treatment: *"A woman who encountered GBV... doesn't get health care because of stigma" (Physical violence survivor 5).* In displaced contexts, this isolation is more acute.

Cultural conventions reinforce stigma. A key informant stated, *"Our tradition and norms, the name given to survivors, discrimination and stigma, all these prevent them from receiving care" (Key informant 2).* Gossip intensifies survivors' dread of exposure. A sexual violence survivor witnessed: *"If you expose yourself, people say this is the one who was attacked or raped" (Sexual violence survivor 2).* Key Informants also highlighted this, adding, *"They fear other people pointing their fingers at them... It is known that in our culture people claim that she is a GBV survivor" (Key informant 1).*

**Community support mechanisms.** Support for survivors is frequently poor. One victim stated, *"I try to talk to the community about what happened to me, but they just gossip. No one gives any meaningful help" (Sexual violence survivor 1).* Another commented, *"In the camp, there's no one to turn to for help" (Physical violence survivor 2).* However, community-based GBV workers have enhanced access to help: *"Now there are women from the IDP community who speak Amharic and are working to help with GBV issues" (Sexual violence survivor 1).* They promote awareness and link survivors to assistance*: "They help collect information and guide women on reporting GBV cases" (Key informant 4).*

### Institutional level barriers and enablers to accessing GBV Services

**Availability and accessibility of GBV Services.** The presence of NGOs and volunteers in camps plays a key role in tackling GBV and promoting access to care. Survivors noted the help offered by these actors. One survivor described, *"There are volunteers or NGOs in the camp. They urge ladies who endured GBV to go to the health facilities by preserving their secrets" (Sexual violence survivor 1).* These actors are crucial in delivering counselling and facilitating referral paths. A physical violence survivor confirmed, *"I reported to one NGO. They asked me if I had health issues due to the violence or not" (Physical violence survivor 3).*

Referral links and one-stop facilities emerged as important facilitators. One-stop facilities provide comprehensive health, legal, and psychological services. A key informant described the services provided at the centre: *"We offer health care, legal services, and counselling here at the centre. All the services are free" (Key informant 5).* Safe places and recreational activities greatly aided survivors' healing. Another key informant added, *"We have safe zones for girls and women in the camp. When a lady with GBV comes to us, her physical and mental well-being will be kept protected" (Key informant 3).*

**Quality of GBV services.** Delays in receiving treatment were a major obstacle, with survivors reporting delays of up to three months. A victim of sexual violence reported, *"I got services three months after the violence" (Sexual violence survivor 3).* Long lines also discouraged help-seeking. *"We obtain services after a lot of waiting. If we don't want to wait, they tell us to come back tomorrow" (Physical violence survivor 4).* Confidentiality issues further hindered survivors' access to care. Another physical violence survivor confirmed, *"The health clinics don't keep our secrets. They don't have privacy rooms" (Physical violence survivor 5).* Many feared intrusions of privacy by camp security officers. Survivors of sexual violence added, *"Not maintaining secrets begins with the security people in the camp. Survivors are reluctant to reveal their condition because they worry their secrets will be disseminated" (Sexual violence survivor 4).* Service providers also shared the concerns raised by survivors, stating, *"The main obstacle is that survivors are afraid someone else will find out about their situation" (Key informant 3).*

**Financial barriers and support.** Although healthcare, legal, and counselling services are free of cost, secondary expenses like transportation fees are barriers. NGOs in the camp alleviate these barriers by offering financial support. As noted by a key informant, *"We support survivors with 3,000 birrs for transportation and counselling" (Key informant 4).* To

ensure survivors' anonymity, secretive financial support is provided for GBV survivors: *"We don't take them to the service centre with our vehicle since people would identify them as GBV survivors. Instead, we give them money so that they can go by themselves "* (Key informant 3). Despite this, financial limitations impede the wider awareness activities: "*It is hard to create awareness to all. Educators are sometimes hampered by inadequate funds" (Key informant 2).*

## Structural level barriers and enablers to accessing GBV Services

**Camp conditions and security challenges.** Lifestyle of camping, as well as security concerns, are serious impediments for GBV survivors in seeking help. Fear of personal safety when seeking health care is a serious impediment. A survivor of GBV testified, "*They do not go to the health facilities because they will encounter a lot of challenges along the way. It is not right for them to visit there [where they receive services] because we are in fear for ourselves. We are in fear." (Sexual violence survivor 3).* A key informant expressed this anxiety, *"There is a security problem because of the current issue in our country" (Key informant 2).*

Overcrowding of the camp drastically affects access to GBV services. Shared living circumstances compromise privacy, exposing survivors to criticism and gossip. One survivor added, *"We are living together... even the female and male sleep together. It would be good if there was a distinct class" (Survivor of physical violence 11).* Another participant reported a lack of privacy in the camp owing to overcrowding, "*People in the camp live close together, so if there is any GBV violence, everyone will surely hear about it. Residents share information about what's going on in the camp. As a result, she may conceal herself." (Physical violence survivor 2).*

**Trust and accountability in the justice system.** Survivors and key informants reported scepticism of the justice system, which they perceived as ineffective and unfair. Many survivors assume that reporting an event will have no serious effects. As one survivor noted, *"It is grab and release; there is no solution" (Physical violence survivor 9).* A GBV service provider added, "*Perpetrators might be arrested but sentenced to only a few years in prison" (Key informant 3).*

Delays in the legal response also tend to discourage survivors. According to one key informant, *"Survivors seeking legal services often encounter slow and unresponsive procedures" (Key informant 5).* Insufficient legal enforcement exacerbates the issue. "*The legal system faces challenges in holding perpetrators accountable, and the process is still developing" (Key informant 2).*

## Discussions

### Prevalence of GBV

In this study, nearly a third of internally displaced women reported at least one form of GBV in their lifetime. This figure is lower than the prevalence of GBV among refugees and IDPs in Africa (48.2%) [5]. The 12-month prevalence of GBV is also lower than that reported among refugees in Benishangul-Gumuz (50.5%) [34], South Sudan (48.3%) [35] and Northwest Ethiopia (37.9%) [4]. This gap might be due to variations in study populations and study tools. First, our study focused only on IDPs in peri-urban camps. Previous research has shown that a higher prevalence of GBV is often reported among studies including refugees than IDPs [5,34,35]. Refugees may have higher rates of GBV than IDPs owing to border crossings, lack of legal protection, and insecure living situations. These issues may be less serious for IDPs in Debre Berhan. Second, the ASIST-GBV tool may underestimate GBV prevalence in our research because it excludes several kinds of violence, such as FGM, kidnapping, and early marriage.

Even though the lifetime GBV prevalence in our research is relatively low, the 12-month prevalence and GBV that occurred in the camps are alarmingly high, with nearly 80% of GBV occurring in the camps. This shows increased vulnerability in IDP camps due to overpopulation, a lack of privacy, poor security, and restricted access to care [5,21,36].

Our results reveal that internally displaced women face various types of violence throughout their displacement journey, including at home, during travel, and in camps, emphasising their ongoing vulnerability. This continuum of violence

throughout their displacement journey underlines the need to provide comprehensive assistance throughout all stages of displacement. Domestic violence (threats, physical violence, forced sex) existed in their home country and increased in camps, whereas opportunistic violence (forced sex, physical abuse) occurred during transportation, similar to prior findings [25,27]. The most common form of GBV in the year before the interview was the threat of violence. Although this form of violence does not always cause physical harm, it may significantly affect survivors' mental health, causing persistent worry, anxiety, and emotional stress.

GBV perpetrators ranged from personal partners to family members, police officers, and the military. More than half of the threats, physical abuse, and forced sex were committed by intimate partners or family members. This is consistent with Dahie et al.'s results, which found that close partners or relatives were responsible for 57.2% of all violence [21]. However, there were reports of GBVs perpetrated by police and soldiers. Similar results were reported in Somalia, where women were subjected to abuse from both strangers and police [37].

GBV survivors suffer exceptionally harsh and dreadful situations when the perpetrators of the violence are family members or others in positions of authority. Those women may suffer intense emotional suffering, fear of revenge, and they may depend on the perpetrators for food, shelter and clothing. All these conditions may, in turn, result in silent, delayed exposure of their GBV status and cause prolonged suffering [33,38,39].

Tailored interventions are needed to address GBV from family members or people who are in a position of power. Some of the methods of doing this involve establishing confidential and anonymous reporting channels for GBV survivors. This can allow GBV survivors to seek assistance without stigma and revenge, especially when the perpetrator is a family member or someone in authority [40]. Another intervention could be equipping counselling and legal aid services with the necessary infrastructure (including trauma-informed services) to handle GBV cases from families or people in a position of power [41]. Lastly, providing sensitisation training for people in a position of power, such as soldiers and police, is important to minimise GBV and promote accountability [42].

## Facilitators of GBV services

This study identified facilitators at the individual, community, and institutional levels that enabled GBV survivors' access to support and care, with participants emphasising the critical necessity of institutional help.

At the institutional level, NGOs' advice was important in expanding access to GBV services. Prior studies highlight the usefulness of NGOs' engagement in improving GBV care. For example, in Dadaab refugee camps, trained refugee community workers, assisted by NGOs, greatly increased survivors' access to services via a comprehensive case management approach [43]. Our study participants also remarked that NGOs performed awareness campaigns, facilitating quick access to treatment. These efforts replicate results from Dadaab, where awareness campaigns boosted help-seeking behaviour among displaced women [23].

One-stop centres (OSCs) were considered major facilitators, combining medical, legal, and psychological help in one place. Research findings demonstrate that OSCs lower obstacles to obtaining GBV services and enhance overall results, including rapid medical treatment and legal paths for justice [44,45].

Safe places for women and girls were another crucial resource, providing survivors with quiet surroundings for healing. Studies underline that these venues provide personalised medical, psychological, and social assistance. For instance, the United Nations Women's Initiative showed that safe places have a positive contribution during emergencies like the COVID-19 pandemic [46]. The Spotlight initiative also stresses their relevance in developing resilience and economic independence [47].

Self-efficacy was recognised as a major facilitator of access to GBV services at the individual level. Research in humanitarian circumstances reveals that women with better self-efficacy and perception of GBV severity are more likely to seek treatment [23]. Self-efficacy improves rehabilitation by strengthening problem-solving abilities and lowering psychological obstacles such as powerlessness, self-blame, and stigma [48,49].

At the community level, community-based GBV workers were crucial in bridging gaps between survivors and services by increasing awareness and giving advice. O'Connell et al. reported that trained community health volunteers successfully disseminated information about SRH services to displaced communities in Ethiopia's Somali region [24]. Muuo et al also reported that although refugee community workers supported service-seeking behaviour, survivors complained about privacy concerns [23]. Addressing this involves educating community workers to give women-centred, culturally sensitive, and confidential care.

## Barriers to access to GBV services

GBV survivors and key informants reported multiple barriers at the different levels of the socio-ecological framework. At the individual level, emotional and psychological responses such as fear of societal stigma, family breakup, self-blame, and other violence from the offender hinder GBV survivors from seeking help. This aligns with the extensive research on humanitarian contexts [21,23,25,50]. These emotional and psychological responses discourage survivors from seeking help and revealing their GBV experiences. Sometimes, GBV survivors withdraw and develop negative coping techniques, which further postpone care.

In addition, insufficient awareness about their rights and the available GBV services exacerbates these hurdles, with some GBV survivors normalising specific sorts of GBV violence. This finding is in line with prior study findings [23,26]. Raising awareness via community campaigns, including trustworthy community leaders, and providing GBV survivors with information on their rights are critical to overcoming these obstacles, especially in humanitarian situations [51,52].

At the communal level, unfavourable attitudes and harmful norms limit survivors' access to care. Certain sorts of GBV, such as FGM and marital abuse, were considered normal by the culture. Negative attitudes towards GBV survivors, such as humiliation and victim blaming, were also recognised as important impediments. Such sorts of norms and unfavourable attitudes led to social isolation and stigmatisation of the victim, further limiting survivors from accessing help and care. The same findings have been found in refugee environments, where some types of abuse, including early marriage, forced sex and FGM, were deemed normal [24,25]. Community rumours about GBV survivors intensify stigma and social isolation, suggesting a need for measures that challenge these detrimental norms and build social support networks [21,24,25].

At the institutional level, loss of anonymity and delay in receiving treatment were the primary problems that inhibited survivors of GBV from seeking care. The limited awareness-raising initiatives and a lack of funding exacerbated these challenges. Similar to our findings, the same barriers have been found in other humanitarian situations, where inadequate care quality and a lack of anonymity make it difficult for GBV survivors to receive care [23,53].

Security problems within the camp and when travelling to the health facilities for treatments were noted as important structural hurdles. Survivors of GBV often refuse to seek assistance and care. Overcrowding and common living arrangements in the camp increase survivors' vulnerability to community gossip, isolation and violation of privacy. Similar findings have been reported elsewhere [21,23,54].

Participants perceived the judicial system as ineffective and inequitable, leading to a loss of trust in legal institutions. Inadequate enforcement and a lack of responsibility further worsen these difficulties, deterring survivors from pursuing justice. Their opinions of the judicial system are consistent with previous studies emphasising the need to restore trust in the legal system and supporting survivor-centred measures [21,24,25]. These strategies are crucial for building trust, ensuring fairness, and pushing survivors to seek justice and assistance.

Despite our study providing important insights on the prevalence of GBV and GBV survivors' access to care, the findings may not be generalizable to all IDPs in Ethiopia and other similar countries. This study was conducted in peri-urban IDP camps, which are near the zone's capital city. As a result, GBV survivors may have better access to support and care compared to remote IDP camps. IDPs in more remote and rural areas may have different GBV experiences due to socioeconomic differences and access-related challenges. Despite this, the findings of this study could be useful for understanding the prevalence of GBV and survivors' access to care in similar peri-urban IDP settings in Ethiopia and other countries.

## Limitations of the Study

This study has certain limitations. First, the ASIST-GBV tool may not include all types of GBVs, such as female genital mutilation, early marriage and abduction. This may underestimate the real prevalence of GBV among IDPs in Debre Berhan. Second, whereas the tool contains a single question that helps to screen threats of violence, it doesn't thoroughly examine psychological acts of violence such as verbal humiliation, manipulation, intimidation or controlling behaviours. This led to the under-reporting of these psychological violence events by GBV survivors.

The participants for the qualitative component were chosen using purposive sampling techniques. This may restrict the generalisability of the qualitative research results. We have implemented numerous measures to minimise this selection bias, such as the inclusion of women with varied age ranges, education level, both physical and sexual violence survivors and key informants with different experiences. To further mitigate selection bias, future research should use more inclusive samples, including targeting marginalised women, including those with disabilities, and adolescents.

## Conclusions and recommendations

GBV is widespread among internally displaced women, especially in camps. Most cases are committed by family members and intimate partners. Survivors face multiple barriers across the socio-ecological framework. Individual barriers include fear of gossip and revenge, lack of knowledge, refusal to disclose their GBV status, and self-isolation. At the community level, negative social norms, stigma, and lack of support increase survivors' silence and hinder access to care. Institutional barriers such as treatment delays, lack of funds, and confidentiality concerns further deter survivors from seeking help. Structural issues like overcrowded camps, instability, and distrust in the legal system significantly limit access to care. Despite these multi-level barriers, facilitators have also been identified. Survivors' self-efficacy at the individual level is vital for help-seeking. Community-based GBV workers promote support and outreach at the community level. Cooperation from NGOs and integrated care at one-stop centres facilitates easier access to care at the institutional level. Implementing coordinated and comprehensive strategies is essential to sustain facilitators and overcome barriers. The following stakeholder-specific recommendations aim to reduce GBV's high burden in IDP camps and enable survivors to access care and support safely and confidently.

### For camp managers and local authorities

The local authorities and managers of the camps should take steps to improve privacy, security, and limit overcrowding within the camps. Along with humanitarian organisations and NGOs, they should upgrade the design, increase the size, and refurbish camps. This will improve safety, increase privacy, and deter overcrowding within the camps.

### For policymakers

Policymakers responsible for GBV prevention and response should enhance the accessibility of the existing one-stop GBV centre by ensuring safe, affordable, and confidential transportation and referral mechanisms for survivors. This not only enables GBV survivors to access integrated health, psychosocial, legal, and protective support in one location but also addresses the privacy concerns raised by survivors. Programmers and policymakers should also foster close partnerships with law enforcement and judiciary agencies to provide survivor-centred legal procedures and protection mechanisms for survivors.

### For NGOs and humanitarian organisations

NGOs and humanitarian organisations should provide a targeted, culturally appropriate sensitisation campaign regarding GBV, available resources, and survivor rights. They should strive to promote safe access to GBV services by covering transportation expenses and addressing safety concerns. NGOs and humanitarian organisations may consider minimising transport expenses by providing subsidies or travel vouchers. They might also enhance security by offering protection training and assisting community-based security initiatives.

## Supporting information

**S1 File. English Version Questionnaire.**
(DOCX)

**S2 File. In-depth interview guideline for GBV survivors.**
(DOCX)

**S3 File. In-depth interview guideline for Key informants.**
(DOCX)

**S4 File. Final modified code-book.**
(DOCX)

**S5 File. The CORQR checklist.**
(DOCX)

**S6 File. GBV SPSS file.**
(CSV)

## Acknowledgments

We are grateful to the research participants, data collectors, and supervisors for their valuable efforts. Our heartfelt gratitude also goes to the coordinators of internally displaced people for supplying vital background information that aided this project.

## Author contributions

**Conceptualization:** Kindie Kebede, Sisay Shiwasinad, Solomon Shiferaw.

**Data curation:** Kindie Kebede, Sisay Shiwasinad, Solomon Shiferaw.

**Formal analysis:** Kindie Kebede, Sisay Shiwasinad, Solomon Shiferaw.

**Funding acquisition:** Kindie Kebede.

**Investigation:** Kindie Kebede, Sisay Shiwasinad, Solomon Shiferaw.

**Methodology:** Kindie Kebede, Sisay Shiwasinad, Solomon Shiferaw.

**Project administration:** Kindie Kebede.

**Resources:** Kindie Kebede, Sisay Shiwasinad, Solomon Shiferaw.

**Supervision:** Kindie Kebede, Solomon Shiferaw.

**Validation:** Sisay Shiwasinad, Solomon Shiferaw.

**Visualization:** Sisay Shiwasinad, Solomon Shiferaw.

**Writing – original draft:** Kindie Kebede, Sisay Shiwasinad, Solomon Shiferaw.

**Writing – review & editing:** Sisay Shiwasinad, Solomon Shiferaw.

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
