## [Decision Letter · Decision Letter 0]

22 Apr 2025

Dear Dr. Kebede,

Thank you for submitting your manuscript to PLOS ONE. After careful consideration, we feel that it has merit but does not fully meet PLOS ONE’s publication criteria as it currently stands. Therefore, we invite you to submit a revised version of the manuscript that addresses the points raised during the review process.

Following my review of your manuscript and the feedback from the reviewers, I would like to emphasize several key points that you should address to improve the rigor and overall clarity of your study:

1. Please include 95% confidence intervals (CIs) for your prevalence estimates, as this will significantly improve the reliability of your findings. A summary table of these estimates, rather than relying solely on visual representations, would also be beneficial.

2. The background section would benefit from a clearer explanation of why Debre Birhan has been a significant location for internally displaced persons (IDPs). Expanding on the socio-cultural, economic, and political factors that contribute to GBV risks will strengthen your justification for focusing on this area.

3. It is crucial to detail the ASIST-GBV questionnaire, including its structure, item types, and psychometric validation within the Ethiopian context. Specifying how qualitative participants were identified and the number of survivors versus key informants will add depth to your methodology.

4. Discuss how your findings may or may not extend to other IDP populations, sharing insights on any sociocultural factors unique to Debre Birhan that could influence GBV rates. Acknowledging potential selection biases in your qualitative component will also provide a clearer framework for interpreting your data.

5. Your conclusion should offer specific actionable recommendations for varied stakeholders, such as NGOs and policymakers, rather than generalized suggestions. This specificity will enhance the practical implications of your findings.

6. Use precise language throughout, especially when discussing the experiences of GBV survivors. Avoid terms that might imply expertise in trauma, and ensure all statements made in the discussion are supported by your data or cautionary language when required.

My additional recommendations and observations can be found below.

We look forward to receiving your revised manuscript.

Kind regards,

Yordanis Enríquez Canto, Ph.D.

Academic Editor

PLOS ONE

Journal Requirements:

“We express our heartfelt appreciation to the consortium of the Guttmacher Institute, Addis Ababa University's School of Public Health, and St. Paul IRHR for their significant support  and funding of this project. We are grateful to the research participants, data collectors, and supervisors for their valuable efforts. Our heartfelt gratitude also goes to the coordinators of internally displaced people for supplying vital background information that aided this project.”

“Authors who received each award: KMK

Grant numbers awarded the author:09/2024

The full name of each funder's. ST.PAUL INSTITUTE FOR REPRODUCTIVE HEALTH AND RIGHTS

URL of each funder website: https://www.spirhr.org/

Funders role: The funder played a role in the study design, data collection, and analysis but did not provide funding for manuscript preparation and publication”

5. Please include a copy of Table 3 which you refer to in your text on page 13.

6. We note you have included a table to which you do not refer in the text of your manuscript. Please ensure that you refer to Table 4 in your text; if accepted, production will need this reference to link the reader to the Table.

**Additional Editor Comments:**

To further strengthen this valuable contribution, we recommend addressing several key points:

Inclusion of 95% Confidence Intervals: We recommend presenting 95% confidence intervals (CIs) for the prevalence estimates of GBV. Including quantitative measures of uncertainty will enhance the reliability and precision of your findings. In addition, it would be beneficial to include a table summarizing these estimates with their corresponding CIs, rather than relying solely on pie charts.

ASIST-GBV Questionnaire Details: Please provide more comprehensive information about the ASIST-GBV instrument used for quantitative data collection. This should include details on its structure, the number and type of items, the domains measured, scoring methodology, and psychometric properties.

Addressing Selection Bias: While your study provides valuable insights through its mixed-methods design, the qualitative component may be subject to selection bias due to the purposive sampling approach for recruiting GBV survivors and NGO staff. We suggest that you discuss this potential bias in more detail, outlining its impact on your findings and considering strategies for mitigating such bias in future research.

Expanding the Discussion: It would strengthen your paper to address how the findings may or may not extend to other internally displaced populations in Ethiopia and in similar settings elsewhere. Discussing factors such as sociocultural, political, and environmental differences will help the reader better understand the broader applicability and any limitations to generalizability.

Alignment of Claims with Data:

Verifying Specific Phrases: We noted certain statements in the discussion and conclusions that warrant more cautious interpretation or further evidence. For example:

The phrase “Survivors confront several interlinked constraints at all levels of the socio-ecological framework” suggests quantifiable interconnections between barriers, although your data primarily offer qualitative insights without statistical quantification. Please either support this statement with additional evidence or rephrase it more cautiously.

Strengthening the connection between all such claims and the data presented or adjusting the language to reflect the exploratory nature of these findings, will improve the credibility and precision of your conclusions.

Stakeholder-Specific Recommendations: Your conclusion would be strengthened by making recommendations more specific and actionable for different stakeholders. Rather than general suggestions, please provide distinct recommendations for policymakers, NGOs, and camp administrators.

Reviewers' comments:

Reviewer's Responses to Questions

**Comments to the Author**

1. Is the manuscript technically sound, and do the data support the conclusions?

Reviewer #1: Yes

Reviewer #2: Yes

Reviewer #3: Yes

2. Has the statistical analysis been performed appropriately and rigorously?

Reviewer #1: Yes

Reviewer #2: Yes

Reviewer #3: Yes

3. Have the authors made all data underlying the findings in their manuscript fully available?

Reviewer #1: No

Reviewer #2: Yes

Reviewer #3: No

4. Is the manuscript presented in an intelligible fashion and written in standard English?

Reviewer #1: Yes

Reviewer #2: Yes

Reviewer #3: Yes

Reviewer #1: Thank you for conducting this important and thoughtful study! I enjoyed reading it. I just have minor comments throughout to help improve

Throughout, just minor formatting issues.

Background

• First line, clarify that GBV does not just affect displaced people, but disproportionately does.

• Second paragraph, spell out what IDP is here, instead of in third paragraph where you do now. I would also have a sentence or two defining it. As it is a key point to your article, you want to make sure everyone is on the same page about what it is.

• Third paragraph, clarify what ‘internal disputes’ are

• Paragraph five, put GBV before SRH as GBV is the focus of the article

Methods

• Is this really mixed methods if you don’t integrate the findings somehow? I would think more multiple method

• I would put study participants before sample size

• More information about how qualitative participants were identified

• Of the 16 participants, how many are survivors and how many are key informants

• Was the questionnaire changed at all after pretesting?

• How long was the questionnaire, how long did it take?

• What else was included in the questionnaire besides measuring GBV?

• How was the ASIST-GBV questionnaire validated in Ethiopia? Include reference, or if it was your team, include details.

• I’d include the six ASIST-GBV in your description, or make a table

• Was the same interview guide used for survivors and key informants?

Data Analysis

• Break up qualitative and quantitative methods more clearly

Ethical Approval

• Do you mean ‘to ensure no harm was done to GBV SUVIVORS, not sufferers?

Results

• Define LARC

• What all is in the questionnaire? I feel like there needs to be more description of the questionnaire but since I don’t know what it included, it’s hard to know (please provide questionnaire as supplementary material)

• Have consistency in how you label the participants after quotes. Sometimes you capitalize each word, other times you don’t

• Shouldn’t structural level come after community level in presentation of result?

Discussion

• Prevalence – you talk about alcohol drinking among IDPs in northern Ethiopia, is that in general or among individuals with GBV? Do you have information on alcohol in your sample? I want to make sure it doesn’t come off as victim blaming

• I’d like you to discuss the complication when the perpetrator is family or a person in authority – that really complicates so many aspects and I believe requires specific interventions.

• I’d like to see more in recommendations – if so much occurs in camps, what specific strategies may be more effective there? Are there examples of initiatives in other locations?

• Also, majority of violence was threatened – what does that really mean? Does that often lead to actual violence? How does that affect stress and well being?

• I’d like to see more in recommendations – if so much occurs in camps, what specific strategies may be more effective there? Are there examples of initiatives in other locations?

Limitations

• What is FGM?

• I wouldn’t say age range was a limitation since that was how you designed your study for reproductive age

Sociecological framework – some of the institutions and health system level indicators don’t have a – or + sign

Reviewer #2: - It is suggested to verify that the keywords are registered as MeSH

- It is necessary to briefly define in the background what is meant by gender-based violence in the context of displaced women.

- In methods, it would be pertinent to add a brief explanation about the socio-ecological framework and how its four levels were interpreted in the study variables.

Reviewer #3: Thank you for the opportunity to review this manuscript on gender-based violence among internally displaced women in Debre Birhan, Ethiopia. The topic addresses an important public health issue affecting displaced women, and I appreciate being able to provide feedback on this timely research. After careful consideration of the paper, I have outlined several strengths and areas for improvement that I believe will enhance the clarity, rigor, and impact of your work. Please find my detailed comments below:

Abstract- Author writes: “...unwillingness to reveal their status" - clarify what "status" refers to in this context. Immigration status? Status as a GBV victim?

Background:

The background section would benefit from a brief explanation of why Debre Birhan specifically has attracted so many IDPs compared to other locations.

Lines 82-84: The sentence "This rising IDP population in Debre Birhan may exacerbate vulnerabilities among women..." uses causal language ("may exacerbate") without establishing the evidence base for this claim. Consider citing previous research demonstrating this relationship.

I would recommend adding a more compelling case for why Debre Birhan specifically deserves research focus compared to other IDP hubs in Ethiopia and/or the African continent. Extant literature exists on GBV in IDP women on the African continent, so what make the current context of focus unique? Essentially, the background mentions Debre Birhan's location and IDP numbers but doesn't establish what makes this setting particularly significant or unique for studying GBV among displaced populations.

Consider addressing some of these points to strengthen your rationale:

1. Explain any distinctive demographic, cultural, or socioeconomic characteristics of Debre Birhan that might influence GBV patterns compared to other IDP settings.

2. Highlight whether Debre Birhan faces unique resource constraints that could exacerbate GBV risks.

3. Note if there are specific governance structures, security arrangements, or service provision models in the Debre Birhan camps that warrant examination.

4. Mention any preliminary observations or previous studies suggesting that GBV patterns in Debre Birhan might differ from national/continental/global trends.

5. Clarify whether Debre Birhan represents an understudied region in comparison to more frequently researched IDP settings in Ethiopia.

Explaining why this specific location presents a compelling case study will greatly strengthen the rationale for your research focus and help readers understand how findings from Debre Birhan might contribute to broader knowledge about GBV in displacement contexts.

Line 106-107: The author writes: “but there is a lack of studies applying this approach to GBV services among IDPs in Ethiopia”. Which studies do exist and what are their findings? An overview of your literature review should be included in the background section to help map out the gaps in the extant literature and the significance of your work.

Methods:

Line 137: How was the census conducted?

Line 157: What type of items did the “structured questionnaire” consist of?

Line 189-191: Elaborate on the processing of validating the questionnaire. How did the author(s) come to the conclusion that the tool had great psychological properties and good reliability in the Ethiopian context?

Discussion

Line 425: The phrase "…are more experienced in GBV than IDPs" is problematic when discussing trauma and violence - it suggests expertise rather than victimization. Instead, use precise language such as "higher prevalence rates of GBV" or "greater exposure to GBV."

Lines 524-25: The potential of the ASIST-GBV tool resulting in underestimation of GBV should be included in the discussion of plausible reasons for why the prevalence of GBV was lower in the sample compared to previous similar studies (line 420-29).

Several statements make causal assertions without sufficient evidence. For example, lines 427-429 attribute higher GBV prevalence in Northern Ethiopia to "sociocultural norms, a lack of social support, and alcohol drinking" without explaining the evidence base for these causal relationships.

The findings from the current study do not seem particularly novel. The discussion would benefit from a stronger emphasis on the specific unique contributions of this study to the field.

**Do you want your identity to be public for this peer review?** For information about this choice, including consent withdrawal, please see our Privacy Policy

Reviewer #1: No

Reviewer #2: No

Reviewer #3: No

---

## [Author Response · Author response to Decision Letter 1]

9 Jun 2025

Response to the editors' comments

1 Please include 95% confidence intervals (CIs) for your prevalence estimates, as this will significantly improve the reliability of your findings. A summary table of these estimates, rather than relying solely on visual representations, would also be beneficial

Response: Thank you for providing this important remark. We incorporated 95% confidence intervals for prevalence estimates, as well as tables to display prevalence values with CIs rather than pie charts and graphs (Result section, lines 724-813)

2 The background section would benefit from a clearer explanation of why Debre Birhan has been a significant location for internally displaced persons (IDPs). Expanding on the socio-cultural, economic, and political factors that contribute to GBV risks will strengthen your justification for focusing on this area.

Response: Thank you for your critical remark in this area. In the updated text, we explain why our research region has been an important destination for internally displaced people (see paragraph 5). We've also added paragraphs on the socio-cultural, economic, and political aspects that contribute to GBV hazards in these camps (please see paragraph 6 of the amended text) (Background section, line 248-257)

3 It is crucial to detail the ASIST-GBV questionnaire, including its structure, item types, and psychometric validation within the Ethiopian context. Specifying how qualitative participants were identified and the number of survivors versus key informants will add depth to your methodology.

Response: We appreciated this thoughtful input. We have given a detailed explanation of the ASIST-GBV questionnaire's structure, contents/domains, and validation in Ethiopia among refugees. Based on the reviewers' comments, we have also included a table displaying the questionnaire. Furthermore, we have included information on how qualitative participants were identified. The number of survivors vs key informants is also included in the revised manuscript (Method section, lines 520-240)

4 Addressing Selection Bias: While your study provides valuable insights through its mixed-methods design, the qualitative component may be subject to selection bias due to the purposive sampling approach for recruiting GBV survivors and NGO staff. We suggest that you discuss this potential bias in more detail, outlining its impact on your findings and considering strategies for mitigating such bias in future research.

Response: Thank you for making this important remark. We have added a note detailing selection biases, their implications, the tactics we used, and future advice for researchers (In the limitation section, lines 1342-1348)

5 Expanding the Discussion: It would strengthen your paper to address how the findings may or may not extend to other internally displaced populations in Ethiopia and in similar settings elsewhere. Discussing factors such as sociocultural, political, and environmental differences will help the reader better understand the broader applicability and any limitations to generalizability.

Response: Thank you for your helpful feedback. We expanded the Discussion section to show how findings may differ across diverse displacement circumstances, highlighting contextual concerns such as social, political, and environmental changes that may limit generalisability (Discussion section, lines 1317-1324)

6 Verifying Specific Phrases: We noted certain statements in the discussion and conclusions that warrant more cautious interpretation or further evidence. For example:

The phrase “Survivors confront several interlinked constraints at all levels of the socio-ecological framework” suggests quantifiable interconnections between barriers, although your data primarily offer qualitative insights without statistical quantification. Please either support this statement with additional evidence or rephrase it more cautiously

Response: Thank you again. We rephrased this comment in the Discussion to better represent the qualitative nature of our findings while avoiding indicating quantitative relationships that were not investigated (Conclusion of the abstract section, line 226-228); Discussion section and Conclusion section, line 1351 and 1352)

7 Strengthening the connection between all such claims and the data presented or adjusting the language to reflect the exploratory nature of these findings will improve the credibility and precision of your conclusions.

Response: Thank you for pointing this out. We've changed the language in both the abstract and the conclusion to reflect the exploratory nature of our qualitative data while avoiding overstating the interconnectedness of findings (Abstract and conclusion sections)

8 Stakeholder-Specific Recommendations: Your conclusion would be strengthened by making recommendations more specific and actionable for different stakeholders. Rather than general suggestions, please provide distinct recommendations for policymakers, NGOs, and camp administrators.

Response: Thank you for your valuable feedback; we have included stakeholder-specific recommendations into the modified draft (Conclusion and recommendation section, line 1358-1391)

9 Use precise language throughout, especially when discussing the experiences of GBV survivors. Avoid terms that might imply expertise in trauma, and ensure all statements made in the discussion are supported by your data or cautionary language when required.

Response: We have avoided language that would indicate expertise in trauma. We tried to ensure all comments made in the debate are supported by our data (All results and discussion section)

10 You indicated that you had ethical approval for your study. In your Methods section, please ensure you have also stated whether you obtained consent from parents or guardians of the minors included in the study or whether the research ethics committee or IRB specifically waived the need for their consent.

Response: We have added information related to consent from parents or guardians of the children. For study participants under the age of 18 years, the research ethics committee waived parental or guardian permission. This is because GBV is a delicate matter, and engaging parents or guardians may result in unintentional injuries or hazards to GBV survivors (Method section, lines 516-519)

11 We note that the grant information you provided in the ‘Funding Information’ and ‘Financial Disclosure’ sections does not match.

Response: Fixed as per your remark.

12 Thank you for stating the following in the Acknowledgements Section of your manuscript:

“We express our heartfelt appreciation to the consortium of the Guttmacher Institute, Addis Ababa University's School of Public Health, and St. Paul IRHR for their significant support and funding of this project. We are grateful to the research participants, data collectors, and supervisors for their valuable efforts. Our heartfelt gratitude also goes to the coordinators of internally displaced people for supplying vital background information that aided this project.”

“Authors who received each award: KMK

Grant numbers awarded the author:09/2024

The full name of each funder's. ST.PAUL INSTITUTE FOR REPRODUCTIVE HEALTH AND RIGHTS

URL of each funder website: https://www.spirhr.org/

Funders role: The funder played a role in the study design, data collection, and analysis but did not provide funding for manuscript preparation and publication”

Response: As directed, we have removed the funding information from the Acknowledgements section of our manuscript. Please revise our Funding Statement as below:

Authors who won each award: KMK

Grant numbers received by the author: 09/2024

Complete names of all funders: St. Paul Institute for Reproductive Health and Rights (SPIRHR), in collaboration with the Guttmacher Institute and School of Public Health at Addis Ababa University

Funder's role: The funder supported study design, data collection, and analysis. They did not support manuscript preparation or publication.

13 Please include a copy of Table 3, which you refer to in your text on page 13.

Response: Thank you for your critical insight. We rectified it as per your remark.

14 We note you have included a table to which you do not refer in the text of your manuscript. Please ensure that you refer to Table 4 in your text; if accepted, production will need this reference to link the reader to the Table.

Response: Corrected as per your remark.

Response to the reviewers’ comments (reviewer 1)

Background

1. First line, clarify that GBV does not just affect displaced people, but disproportionately does.

Response: We appreciate this. This is an excellent observation. The criticism has been gratefully acknowledged, and a statement in the second paragraph describes how GBV affects displaced persons differently from non-displaced people. We included this in the second paragraph to preserve the logical flow and consistency of paragraphs, as well as to address other reviewers' concerns (Background section, line 105)

2 Second paragraph, spell out what IDP is here, instead of in third paragraph where you do now. I would also have a sentence or two defining it. As it is a key point to your article, you want to make sure everyone is on the same page about what it is.

Response: We assumed that readers of this text understood what IDPs are. However, as you correctly pointed out, understanding what IDPs are is critical to ensuring that all readers are on the same page. Thank you for your critical observation. In response to your feedback, we have included a definition of IDPS in the third paragraph. We included the term in the third paragraph to preserve the logical flow and cohesion of the paragraphs (Background section, line 112)

3 Third paragraph, clarify what ‘internal disputes’ are

Response: We have replaced the term “internal disputes” with armed conflicts and inter-ethnic clashes (Background section, line 136)

4 Paragraph five, put GBV before SRH, as GBV is the focus of the article

Response: Corrected as per your comment (Background section, line 179)

Method

1 Is this really mixed methods if you don’t integrate the findings somehow? I would think more multiple method

Response: In our study, we used a parallel mixed-methods design with quantitative and qualitative components to address different research objectives: the quantitative component estimated the prevalence of gender-based violence (GBV), while the qualitative component investigated the barriers and facilitators to care access among GBV survivors. Although we presented the data individually based on their respective goals, we integrated the results throughout the interpretation and discussion phase to offer a comprehensive understanding of the scope and context of GBV in our study setting. We have now modified the methodology section to better clarify the justification for using a parallel mixed-methods architecture, as well as how conceptual integration was accomplished in the discussion (Method section, Line 300-305)

2 I would put study participants before sample size

Response: As per your feedback, we have positioned the study participants ahead of the sample size (Method section, lines 306-310)

3 More information about how qualitative participants were identified

Response: Thank you for asking for further information about how qualitative participants were identified. We have provided data regarding the identification of the study participants (Method section, lines 377-387)

4 Of the 16 participants, how many are survivors and how many are key informants

Response: The research included eleven GBV survivors and five key informants. We have incorporated this information in the amended manuscript's sample size section (Method section, line 333)

5 Was the questionnaire changed at all after pretesting?

Response: Thank you for expressing this concern. We made a minor change to the questionnaire following the pretest. We included this information in the data gathering section (Method section, Line 347)

6 How long was the questionnaire, how long did it take?

Response: Thank you for reminding us to mention the time required to complete the survey. We have put a notice on this (please see the Data collecting tools and methods page)(Method section, Lines 349 and 340)

7 What else was included in the questionnaire besides measuring GBV?

Response: The questionnaire covered respondents' sociodemographic information, categories of GBV, when they happened, and the perpetrators of the violence(kindly refer to the Data gathering methods and procedures part (Methods section, lines 348-350)

8 How was the ASIST-GBV questionnaire validated in Ethiopia? Include reference, or if it was your team, include details.

Response: Yes, the instrument was validated in Ethiopia in the refugee situation, and it has been utilised in additional research among internally displaced persons in northern Ethiopia. We have supplied references and a full discussion (Method section, lines 412-450)

9 I’d include the six ASIST-GBV in your description, or make a table

Response: Thank you for reminding us to provide a description or table regarding six ASIST-GBV. we have provided an explanation as well as a table that includes all six questions in Table 1 (Method section, Lines 431 and 432)

10 Was the same interview guide used for survivors and key informants?

Response: We employed different interview instructions for key informants and GBV survivors, while the main contents are practically comparable. We have erroneously provided just one of the interview rules, which is solely for GBV survivors. Sorry for this accidental error and the difficulty we have caused. Now we have incorporated two distinct interview instructions for key informants and GBV survivors in the amended manuscript (Interview guides 2 and 3) (Method section, line 353)

11 Break up qualitative and quantitative methods more clearly

Response: We have divided the qualitative and quantitative techniques for improved clarity (please refer to the analysis section of the modified text)(Method section, lines 433-473)

12 Do you mean ‘to ensure no harm was done to GBV SUVIVORS, not sufferers?

Response: It refers to those who have survived gender-based abuse. I apologise for any misunderstandings. We truly appreciated your comment and corrected the grammatical error(Method section, line 519)

Results

1 Define LARC

Response: The amended paper now includes a comprehensive description of LARC(Results section line 533)

2 What all is in the questionnaire? I feel like there needs to be more description of the questionnaire, but since I don’t know what it included, it’s hard to know (please provide the questionnaire as supplementary material)

Response: Thank you for reminding us to include the questionnaire as supplementary material. We have included in the revised manuscript (see additional file 1, Method section, Line 349)

3 Have consistency in how you label the participants after quotes. Sometimes you capitalize each word, other times you don’t

Response: We have tried to maintain consistency in labelling the participants after quotes throughout the revised document. Thank you for this important editorial comment.

4 Shouldn’t the structural level come after the community level in the presentation of results?

Response: A nice observation. Yes, as you said, and based on our framework. The structural level should come after the community and even after the institutional level. We addressed this remark in the amended text (Result section, line 1066-1097)

Discussion

1 Prevalence – you talk about

---

## [Decision Letter · Decision Letter 1]

1 Jul 2025

Dear Dr. Kebede,

We look forward to receiving your revised manuscript.

Kind regards,

Yordanis Enríquez Canto, Ph.D.

Academic Editor

PLOS ONE

Journal Requirements:

**Additional Editor Comments:**

Reviewer Status Update:

I want to inform you about the review process for this revision. Of the three reviewers originally contacted for this manuscript, one declined the invitation, and we received no response from another reviewer despite multiple attempts to contact them. We were able to obtain a comprehensive review from one reviewer (Reviewer 2), whose feedback has been valuable in assessing your revisions. To avoid unnecessarily delaying the review process while maintaining the quality standards of PLOS One, I am proceeding with the editorial decision based on this single review and my own assessment of the manuscript.

After careful consideration of your revised manuscript and Reviewer 2's comments, I am pleased to inform you that your manuscript is conditionally acceptable for publication, pending minor revisions to address the remaining concerns raised by the reviewer.

Required Revisions:

1. Enhanced Definition of Gender-Based Violence in Displacement Context

While your manuscript provides a general definition of GBV, Reviewer 2 correctly identifies a gap in contextualizing this definition specifically for displaced women. Please expand your definition in the Background section to explicitly address how GBV manifests in displacement contexts. Consider incorporating guidance from authoritative sources such as the UN High Commissioner for Refugees (UNHCR) to provide a more comprehensive definition.

2. Integration of Ecological Framework in Conclusions

Your study effectively employs the socio-ecological framework throughout the methodology and results. However, please strengthen the conclusions section by explicitly referencing this framework when interpreting your findings. Consider:

Organizing your concluding remarks around the four levels of the ecological framework (individual, community, institutional, structural)

Given the minor nature of these revisions, I anticipate being able to make a final decision promptly upon receipt of your revised submission.

Your study makes an important contribution to understanding GBV among internally displaced women and provides valuable insights for policy and practice. The mixed-methods approach and use of the socio-ecological framework strengthen the manuscript significantly.

Reviewers' comments:

Reviewer's Responses to Questions

**Comments to the Author**

Reviewer #2: (No Response)

2. Is the manuscript technically sound, and do the data support the conclusions?

Reviewer #2: Partly

3. Has the statistical analysis been performed appropriately and rigorously?

Reviewer #2: Yes

4. Have the authors made all data underlying the findings in their manuscript fully available?

Reviewer #2: Yes

5. Is the manuscript presented in an intelligible fashion and written in standard English?

Reviewer #2: Yes

Reviewer #2: - Regarding the second observation, there is still a knowledge gap regarding the definition of gender-based violence in the context of displaced women. Although the context of displacement, the consequences for women, and the risk factors that perpetuate gender-based violence have been repeatedly explained, what constitutes gender-based violence in displaced women has not been specified. For example, "it is any physical, psychological, sexual, and economic factor that manifests itself with greater risk in the context of displacement or mobility; therefore, it can occur through sexual exploitation, harassment, abuse of authority, etc." In this regard, it is suggested that the example be used as a guide and that information be sought from expert institutions such as the United Nations High Commissioner for Refugees to add more precise information to the requested definition.

- It is suggested to include the ecological framework to interpret the results in the conclusions.

**Do you want your identity to be public for this peer review?** For information about this choice, including consent withdrawal, please see our Privacy Policy

Reviewer #2: No

---

## [Author Response · Author response to Decision Letter 2]

6 Jul 2025

Response to Reviewers

Manuscript ID: PONE-D-25-11392R1

We appreciate the academic editor and reviewer 2's complimentary and intelligent remarks and suggestions. Below is a point-by-point discussion on how each issue was handled in the final document.

Editor/Reviewer Comment 1: Enhanced Definition of GBV in Displacement Setting

"While your manuscript provides a general definition of GBV, Reviewer 2 correctly identifies a gap in contextualising this definition specifically for displaced women. Please expand your definition in the Background section to explicitly address how GBV manifests in displacement contexts. Consider incorporating guidance from authoritative sources such as the UN High Commissioner for Refugees (UNHCR) to provide a more comprehensive definition.”

Response: We appreciate this important suggestion. We have revised the background section (Paragraph 2, line 77-81) to include a concrete, context-specific definition of GBV among displaced women. The revised paragraph is:

"In the context of displacement, GBV includes any physical, psychological, sexual or economic act that harms women's safety and dignity due to displacement. In displacement situations, GBV can take many forms, such as sexual abuse and exploitation in exchange for basic survival needs, early or forced marriage, denial of access to essential services, violence by intimate partners, state and non-state actors, and workers from humanitarian organisations.”

Editor/Reviewer Comment 2: Ecological Framework Integration in Conclusions

"Your study effectively employs the socio-ecological framework throughout the methodology and results. However, please strengthen the conclusions section by explicitly referencing this framework when interpreting your findings. Consider: Organising your concluding remarks around the four levels of the ecological framework (individual, community, institutional, structural)”

Response: We agree with this conclusion and have reworded the conclusions section accordingly. The last paragraph has been extended to discuss findings within the socio-ecological model, emphasising multi-level facilitators and barriers (kindly see the conclusion section, lines 670- 682).

---

## [Decision Letter · Decision Letter 2]

23 Jul 2025

Investigating Gender-Based Violence Against Internally Displaced Women in Debre Berhan, Central Ethiopia: A Mixed-Methods Study Using the Socio-Ecological Framework

PONE-D-25-11392R2

Dear Dr. Kebede,

We’re pleased to inform you that your manuscript has been judged scientifically suitable for publication and will be formally accepted for publication once it meets all outstanding technical requirements.

Kind regards,

Yordanis Enríquez Canto, Ph.D.

Academic Editor

PLOS ONE

Additional Editor Comments (optional):

Reviewers' comments:

Reviewer's Responses to Questions

**Comments to the Author**

Reviewer #2: All comments have been addressed

2. Is the manuscript technically sound, and do the data support the conclusions?

Reviewer #2: Yes

3. Has the statistical analysis been performed appropriately and rigorously?

Reviewer #2: Yes

4. Have the authors made all data underlying the findings in their manuscript fully available?

Reviewer #2: Yes

5. Is the manuscript presented in an intelligible fashion and written in standard English?

Reviewer #2: Yes

Reviewer #2: (No Response)

**Do you want your identity to be public for this peer review?** For information about this choice, including consent withdrawal, please see our Privacy Policy

Reviewer #2: No

---

## [Editor Report · Acceptance letter]

PONE-D-25-11392R2

PLOS ONE

Dear Dr. Kebede,

I'm pleased to inform you that your manuscript has been deemed suitable for publication in PLOS ONE. Congratulations! Your manuscript is now being handed over to our production team.

Kind regards,

on behalf of

Prof. Yordanis Enríquez Canto

Academic Editor

PLOS ONE